# Anatomical and functional investigation of the marmoset default mode network

Cirong Liu[1], Cecil Chern-Chyi Yen [1], Diego Szczupak[1], Frank Q. Ye [2], David A. Leopold[2,3] & Afonso C. Silva [1]

The default mode network (DMN) is associated with a wide range of brain functions. In humans, the DMN is marked by strong functional connectivity among three core regions: medial prefrontal cortex (mPFC), posterior parietal cortex (PPC), and the medial parietal and posterior cingulate cortex (PCC). Neuroimaging studies have shown that the DMN also exists in non-human primates, suggesting that it may be a conserved feature of the primate brain. Here, we found that, in common marmosets, the dorsolateral prefrontal cortex (dlPFC; peak at A8aD) has robust fMRI functional connectivity and reciprocal anatomical connections with the posterior DMN core regions (PPC and PCC), while the mPFC has weak connections with the posterior DMN core regions. This strong dlPFC but weak mPFC connectivity in marmoset differs markedly from the stereotypical DMN in humans. The mPFC may be involved in brain functions that are further developed in humans than in other primates.

[1] Cerebral Microcirculation Section, Laboratory of Functional and Molecular Imaging, National Institute of Neurological Disorders and Stroke, National Institutes of Health, Bethesda, MD 20892, USA. [2] Neurophysiology Imaging Facility, National Institute of Mental Health, National Institute of Neurological Disorders and Stroke, and National Eye Institute, National Institutes of Health, Bethesda, MD 20892, USA. [3] Section on Cognitive Neurophysiology and Imaging, Laboratory of Neuropsychology, National Institute of Mental Health, National Institutes of Health, Bethesda, MD 20892, USA. Correspondence and requests for materials should be addressed to C.L. (email: cirong.liu@nih.gov) or to A.C.S. (email: afonso@pitt.edu)

The default mode network (DMN) is a collection of cortical areas now widely acknowledged as a principal network component of the brain's functional architecture[1,2]. While its role in cognition is still a matter of debate, disruptions of the DMN have been reported in a wide range of neurological and neuropsychiatric diseases[3]. Although recent advances in human brain mapping have greatly improved characterization of the DMN at the macroscale, the anatomical and physiological basis of the DMN, as well as its mechanistic contribution to brain function, are poorly understood. Hence, there is a need for direct investigation of the DMN in animal models. The first step in this direction is to identify the DMN's areal components. In humans, this is most commonly achieved by mapping spatial patterns of correlated spontaneous fMRI signals obtained in the absence of any overt task, commonly dubbed resting-state fMRI (rsfMRI)[2]. The DMN in humans is marked by three highly correlated core clusters, as well as some less prominent ones. The first core cluster is located anterior-medially within the medial prefrontal cortex (mPFC). The second principal cluster is located posterior-medially and consists of the medial parietal and posterior cingulate and retrosplenial cortices (PCC). The third significant cluster is posterior-lateral and located within the posterior parietal cortex (PPC). Recent studies revealed nominally similar DMN in non-human primates (NHP), including chimpanzees[4] and macaques[5,6]. One shared feature in these species is the anterior-medial component (like the mPFC of humans), exhibiting strong functional connectivity with the PCC and the PPC. This gross anterior–posterior pattern is now commonly regarded as a typical feature of the DMN[1].

The common marmoset (Callithrix jacchus) is a small NHP species with a nearly smooth brain that has gained wide interest due to its promise in biomedical research, including as a model of the functional organization of the primate brain[7–10]. A pioneering study in awake marmosets used independent component analysis (ICA) to assess the intrinsic functional networks of the marmoset brain during rest[7]. This study revealed a DMN component that included areas around the PCC, the PPC, and dorsal–lateral prefrontal cortex (dlPFC). Absent, however, was the other main component of the DMN, the mPFC, calling into question the specific functional role of mPFC and its relevance to the DMN in marmosets.

It is important to note that the DMN was initially identified using positron emission tomography and named after its ostensible deactivation during a wide range of attention-demanding tasks[2]: the brain network was active by default, but then suppressed during a task. Only afterward did correlation-based resting-state fMRI studies revealed that those same task-negative regions are also strongly coupled in their spontaneous hemodynamic fluctuations[2]. Thus, the behavior of candidate DMN areas during task performance may be relevant in understanding the apparent discrepancy between marmosets[7] and the other primate species with reported DMN[4–6].

In this study, we systematically investigated the DMN in marmosets using multiple approaches. First, we replicated our previous study's characterization of the DMN in this species[7] using new and improved resting-state fMRI data. Next, we analyzed task-based fMRI data to examine the extent to which putative DMN regions overlapped with task-negative brain regions and compared results to DMN patterns in macaques and humans. Finally, we investigated anatomical connections of each node of the putative DMN using neuronal tracing data, to determine whether direct anatomical connections support the particulas pattern of spontaneous correlations, as previously reported in macaques[6]. Our findings demonstrate a marked difference in the marmoset compared to humans. Marmosets exhibit a strong lateral, but a weak medial, DMN node in their frontal

cortex, thus deviating from the stereotypical pattern (mPFC–PCC–PPC) of the DMN in humans.

## Results

**DMN regions deactivate during functional tasks.** We collected awake resting-state fMRI data from 7 marmosets for 4–6 runs each (each run was 17 min long) and performed ICA on all subjects (group-ICA) and on each individual (subject-ICA). The ICA extracts coalitions of voxels exhibiting spatiotemporal covariation and reflects the correlated activity of the fMRI signal across the brain. In line with our previous study[7], we identified one ICA component that was termed the DMN (Figs. 1a–c and 2a). The main areas in this ICA component included the dorsolateral prefrontal cortex (dlPFC; peak at area 8aD and rostral 6DR), the medial parietal and posterior cingulate cortex (PCC; peak at areas PGM and caudal A23/A29/A30), and the posterior parietal cortex (PPC; peak at the intraparietal area). These three regions could be robustly detected not only in the group-ICA (Fig. 1a, b) but also in the subject-ICA of every single individual (Fig. 1c), regardless of different settings of the number of components and thresholds. In addition to the three regions specified above, we detected patches in the temporal area (TC) and parahippocampal regions (PHC), but these areas were not as prominent as the three main areas above (Fig. 1a, b). A small patch near mPFC (A32) was also observed as a weak component, but it only survived under the default low threshold and could not be replicated in all animals (Fig. 1c). Taken together, the areas in the ICA component matched well those previously described as belonging to the DMN of the marmoset[7]. Three regions—dlPFC, PPC, and PCC—form the primary spatial pattern of the resting DMN in marmosets.

We next investigated whether the same regions confirm their inclusion in the DMN functionally through task deactivation, as it has been shown previously in humans and macaques[2,6]. We analyzed task-fMRI data in which two marmosets were trained to attend to visual stimuli presented periodically[12] (Fig. 1d). Despite the simplicity of the task, we detected significant deactivation of the PCC, PPC, and PHC (Figs. 1e, f, and 2b) in both animals (stimuli vs. resting). The DMN regions PCC and PPC, determined by ICA of resting-state fMRI data, therefore, also showed negative responses (deactivation) to task-based fMRI. Both PCC and PPC were also the central regions of the DMN in humans and macaques (Fig. 2c, d), and neither region overlapped with regions that showed positive responses (activation) during the task (Fig. 2b). The PHC regions, especially the entorhinal cortex, which significantly deactivated during the task, were weak components in the resting ICA DMN of marmosets, suggesting the existence of different DMN subsystems. We did not detect any significant deactivation of the dlPFC in our task-based fMRI data, which may be due to the simplicity of the visual task, as a possibly homologous dlPFC patch (A8b) was deactivated in all visual tasks in macaques (Fig. 2c). The evidence presented above confirms that the areas in the ICA component are the marmoset DMN, as they mostly overlap with task-negative regions and none of them overlap with task-positive areas.

**dlPFC, not mPFC, is a prominent region of the marmoset DMN.** The mPFC is a prominent core region of the DMN in humans, which occupies a vast region and has robust functional connectivity with the PCC (Fig. 2d). In macaques, although much smaller in size and maybe not homologous to the human mPFC, a similar anterior-medial region (A24/32, Fig. 2c) has robust functional connectivity with the PCC[5] and it can also be detected by deactivation during functional tasks[6]. In marmosets, however, the mPFC appeared to be a weak component of the DMN (Figs. 1

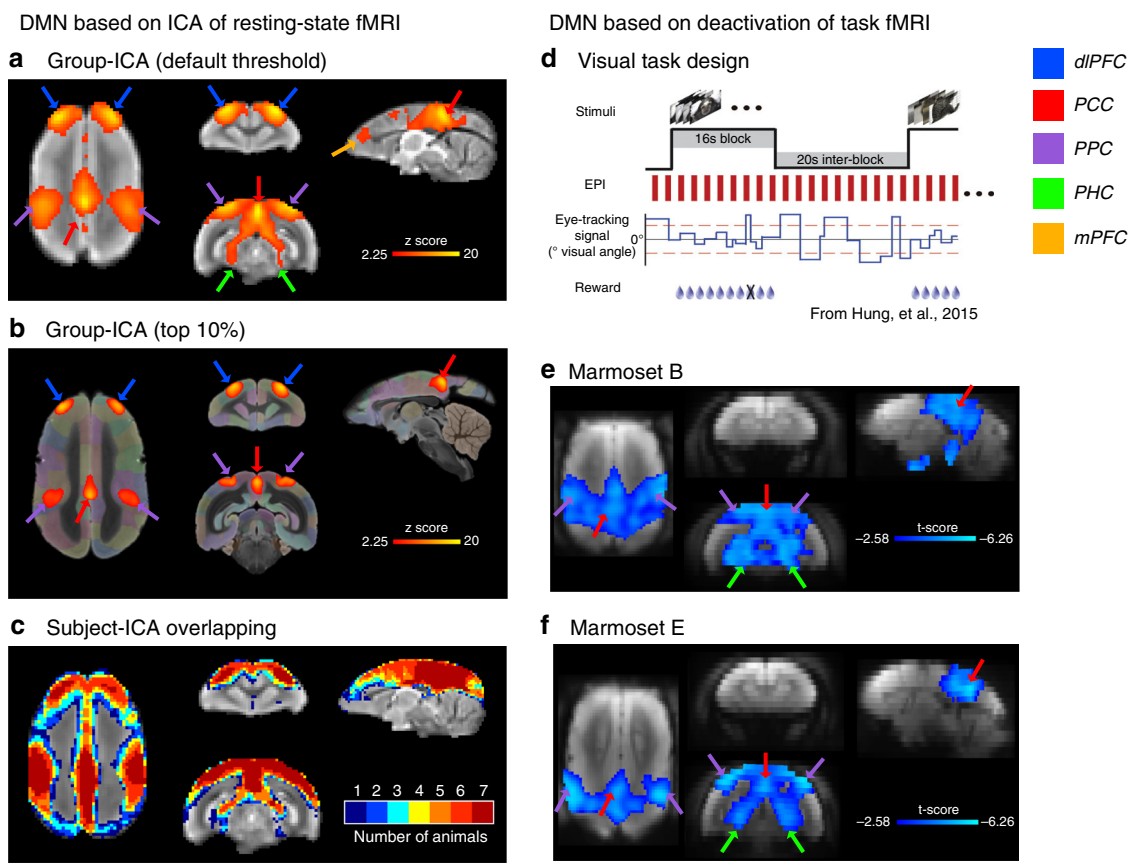

**Fig. 1** Regions of the DMN largely overlap negative responding regions to task-based fMRI. **a** Putative regions of the DMN were extracted from group-wise ICA of resting-state fMRI data across seven marmosets, overlaid on mean EPI images. The ICA component was thresholded at a level of 0.5 using a mixture model and an alternative hypothesis testing approach (default of the FSL-Melodic). Surface rendering of this ICA map is presented in Fig. 2a. **b** Peak regions of the DMN component (the top 10% voxels), overlaid on a template from the NIH Marmoset Brain Atlas[11]. **c** Subject-specific ICA was performed and putative DMN regions were extracted from each marmoset using the default threshold, which are shown in Supplementary Fig. 1. An incidence map of the overlap of these subject-specific ICA components is presented. **d** Block-design paradigm for the visual stimulation task-based fMRI. Two marmosets (**e**, **f**) were trained to attend to visual stimuli presented in 16-s long blocks on an LCD screen, during which different images were randomly presented in each block[12]. The 16-s long blocks were intercalated by 20-s resting blocks during which the screen was blank (gray). Regions that deactivated during the stimulus block relative to the blank screen are presented (stimulus—resting, p < 0.05, Student's t-test, corrected for multiple comparisons). dlPFC dorsal–lateral prefrontal cortex, PPC posterior parietal cortex, PCC posterior cingulate and medial parietal cortex, PHC parahippocampal cortices, mPFC medial prefrontal cortex. Arrows identify the different regions according to the color legend

and 2). We extracted the mean time courses of each area and calculated their functional connectivity. The three main regions of the DMN (the dlPFC, PCC, and PPC) had robust functional connectivity with each other, whereas the mPFC had lower connectivity with the posterior DMN regions (Fig. 3). Thus, a weak mPFC component of the marmoset DMN is the most obvious difference from the general pattern of the human DMN. On the other hand, the dlPFC (A8aD/rostral 6DR) appeared to be an essential region of the marmoset DMN (Figs. 1–3). However, this is not a unique feature of marmosets, because a similar dlPFC patch was also a typical component in human and macaque DMN (Fig. 2c, d), albeit seldom discussed in previous literature.

**Marmoset DMN regions are anatomically connected.** Previous studies suggested that anatomical connections support the robust connectivity among DMN regions[1]. In marmosets, two questions can be raised: (1) Are the three DMN regions (the dlPFC, the PCC, and the PPC) anatomically connected? In particular, the dlPFC was rarely investigated as part of the DMN, and it remains unknown whether dlPFC has anatomical connections to either the PPC, the PCC, or both, to support its robust connectivity. (2)

Is the weak functional connectivity of the mPFC with other DMN regions due to a lack of direct anatomical connections?

To address these questions here, we used a recently published marmoset retrograde tracing database[14]. Three DMN regions had direct (reciprocal and monosynaptic) connections with each other (Fig. 4a–c and Supplementary Fig. 3A-C). The dlPFC was both functionally and anatomically connected with both posterior DMN regions. On the other hand, the mPFC showed a distinct anatomical connectivity pattern, with few direct anatomical connections with other DMN regions (Fig. 4d and Supplementary Fig. 3D), which is consistent with its low correlation with the DMN regions. Injections in the mPFC revealed cell bodies labeled in retrosplenial cortex, area prostriata and PHC regions (Supplementary Fig. 3D), which were also weak components of the marmoset DMN. These results demonstrate that anatomical connections support the functional connectivity of the core DMN regions in the marmoset brain.

**Seed-based analysis confirms results obtained by ICA.** Although the mPFC regions (A32, A32v, 10m) did not project from/to the peak regions of the DMN, they showed anatomical

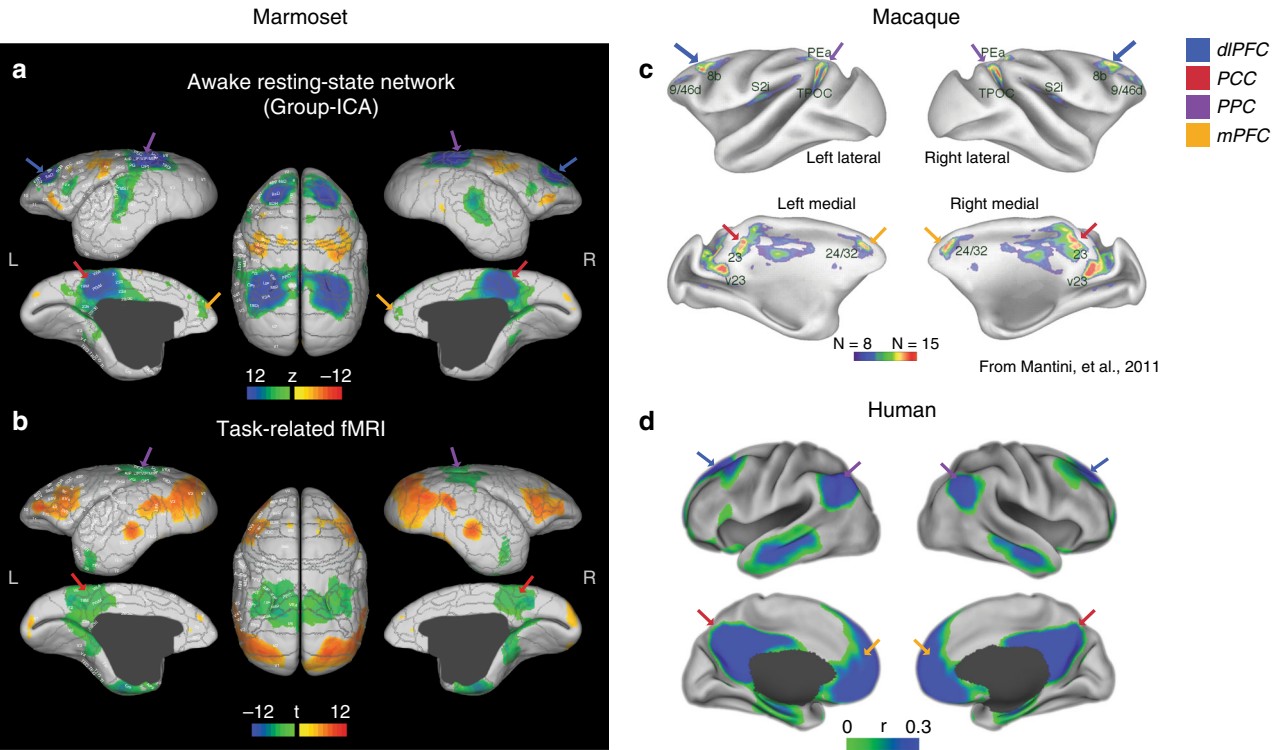

**Fig. 2** The marmoset DMN and cross-species comparison. **a** The group-ICA DMN component of marmosets under the default (low) threshold is shown on a brain surface map. The color map is reversed (i.e., the cold colors represent the DMN regions). Areal borders are the Paxinos parcellation from the NIH Marmoset Brain Atlas[11,15]. **b** Regions that activate (warm colors) and deactivate (cold colors) during the visual task (Fig. 1d) in two marmosets. **c** DMN in macaques adapted from Mantini et al.[6], where regions that deactivate in at least 8 out of 15 different tasks are shown. The color bar shows the number of tasks during which a region deactivates. **d** Typical human DMN generated by the Neurosynth metanalysis[13], which shows the functional connectivity to the posterior cingulate cortex (seeding at the coordinate (0, −44, 28)). Note that atlas labels of marmosets and macaques are defined by different anatomical atlases, which do not have one-to-one homologous relationships across species. dlPFC dorsal–lateral prefrontal cortex, PPC posterior parietal cortex, PCC posterior cingulate and medial parietal cortex, mPFC medial prefrontal cortex. Arrows identify the different regions according to the color legend

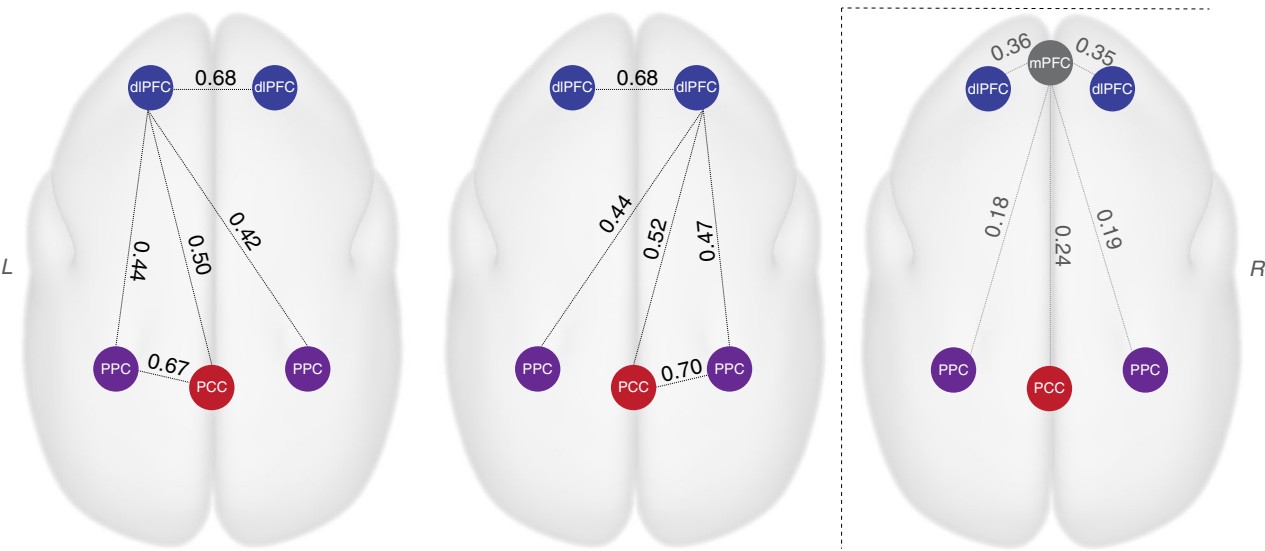

**Fig. 3** Resting-state functional connectivity among each region of the marmoset DMN. The three main regions of the marmoset DMN (dlPFC, PPC, and PCC) have high correlation coefficients with each other, whereas the mPFC is weakly correlated to the other DMN regions. The correlation values here are mean correlations values across all marmosets. The correlations for each marmoset are presented in Supplementary Fig. 2. dlPFC dorsal–lateral prefrontal cortex, PPC posterior parietal cortex, PCC posterior cingulate and medial parietal cortex, mPFC medial prefrontal cortex. R right, L left

Retrograde neuronal tracing

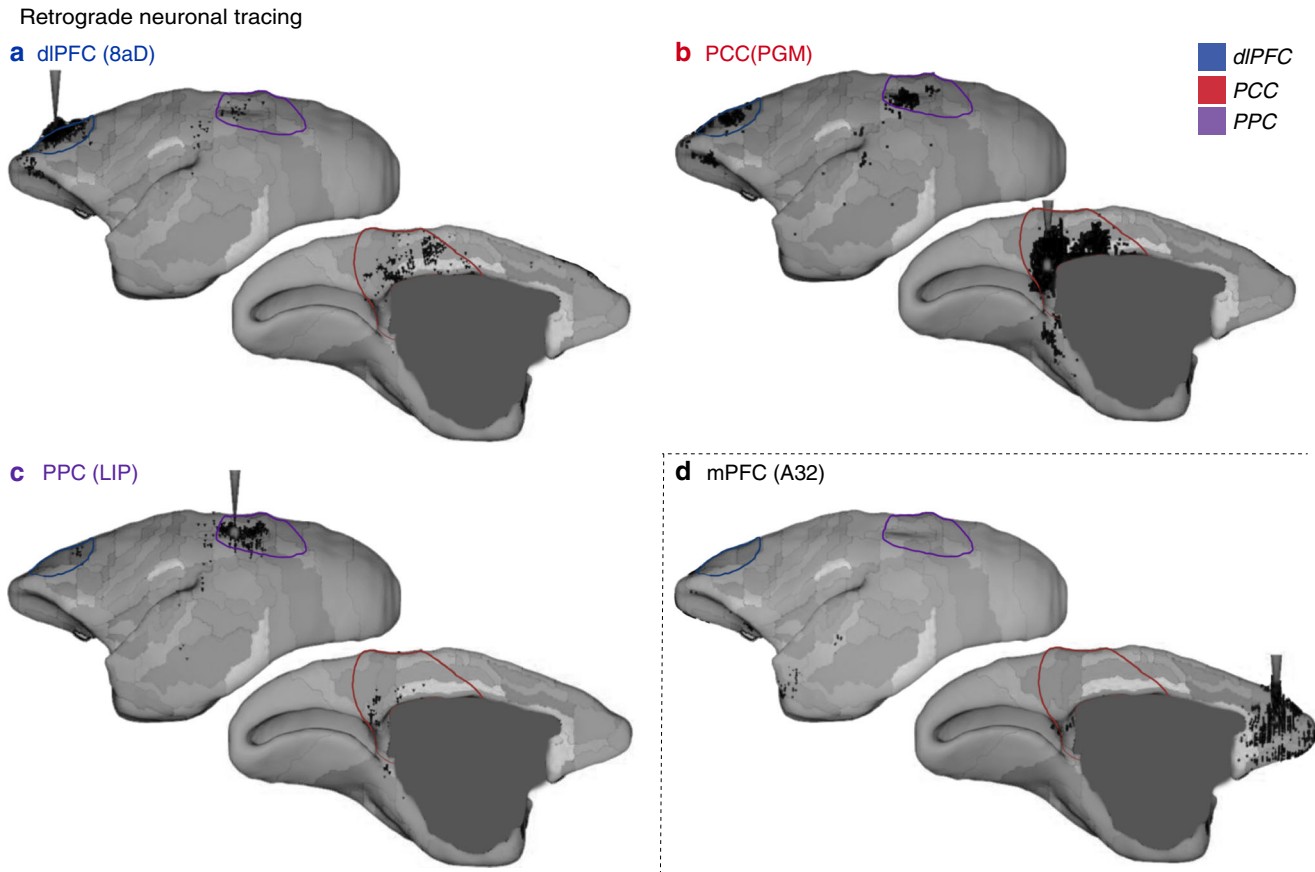

**Fig. 4** DMN regions have direct anatomical connections with each other but not the mPFC. Retrograde tracing data are retrieved from the Monash Marmoset Brain Architecture Project[14]. Colored borders are manually overlaid on the tracing maps for visualization, which represent the approximate location and size of each of the three DMN regions as determined by ICA of resting-state fMRI data using a default threshold (Fig. 1a). The dlPFC is in blue, the PCC in red, and the PPC in purple. Injection cases of CJ108 (**a**), CJ84 (**b**), CJ55 (**c**), and CJ187(**d**) are presented. More injection examples of related areas are presented in Supplementary Fig. 1. Note that areas A8aD, PGM, and LIP are terms that were defined by the Paxinos parcellation[15], which do not necessarily have a one-to-one homology with other species. dlPFC dorsal–lateral prefrontal cortex, PPC posterior parietal cortex, PCC posterior cingulate and medial parietal cortex, mPFC medial prefrontal cortex. Arrows identify the different regions according to the color legend

connections with the retrosplenial cortex, which is part of human DMN. Considering the small size of the frontal pole and mPFC in the marmoset, it was possible that the ICA analysis we performed failed to reveal the "mPFC–PCC" functional connectivity associated with the DMN. To investigate this possibility, we performed an exploratory seed-based analysis of our resting-state fMRI data. We manually defined 47 seeds (Fig. 5a). Nineteen seeds were placed across the mPFC, involving the frontal pole (A10m), A32, A32v, and medial parts of A14R, A9, and A8b. Twenty-eight seeds were placed across the medial parietal, posterior cingulate and retrosplenial cortex, involving A29/30, 23, PGm, 19m, and medial part of V6.

We performed a whole-brain connectivity analysis on each seed. Despite the vast number of seeds across the mPFC (seeds 1–19), none of them showed stronger connectivity with the PCC than those revealed in the previous ICA (Fig. 5 and Supplementary Figs. 4 and 5). The frontal pole (seeds 1–4) had a low correlation with other distant brain regions ($r < 0.15$). Consistently, seeds in the PCC and retrosplenial cortex did not show any strong correlation with the mPFC or the frontal pole either. On the contrary, the connectivity patterns from seeds in the PCC and retrosplenial cortex (seeds 20–47) were similar to the DMN revealed by ICA, with correlation peaked at around the A8aD, the intraparietal area and the seed itself (Fig. 5 and Supplementary Fig. 4). The area within the mPFC that had the highest correlation ($r \sim 0.1$–$0.18$) with seeds placed in PCC was located around A32

(Supplementary Fig. 5), the same region revealed by ICA as a weak component of the DMN (Figs. 1a and 2a). Therefore, the exploratory seed-based analysis confirmed the ICA, as well as the previous studies[7,10], and it confirmed that neither mPFC nor the frontal pole had a strong correlation with the PCC.

## Discussion

The DMN is one of the most prominent resting-state networks in the marmoset brain, and one that can be robustly extracted from resting-state fMRI data[7]. However, the DMN in marmosets differs from that in humans, specifically in the prefrontal cortex. Our data showed that dlPFC is the prominent anterior region of the marmoset DMN. While the mPFC is one of the most evident and significant cores of the human DMN, it is a weak component in the marmoset DMN.

By analyzing individual functional data, a recent study revealed two parallel DMN in humans that possess closely juxtaposed regions in several cortical zones[16]. One human DMN network (DMN-A) involved high correlations in the retrosplenial cortex and PHC, but the other does not (DMN-B). Tracing data showed that, in marmosets, the mPFC and the frontal pole (A10) receive connections from the retrosplenial cortex and PHC (mainly temporal area TH), which are typical DMN-A regions in humans. A more thorough analysis of the marmoset neuronal tracing data (monosynaptic)[14] by Buckner et al.[17] further indicates that the frontopolar region (A10) may be involved in a DMN-like

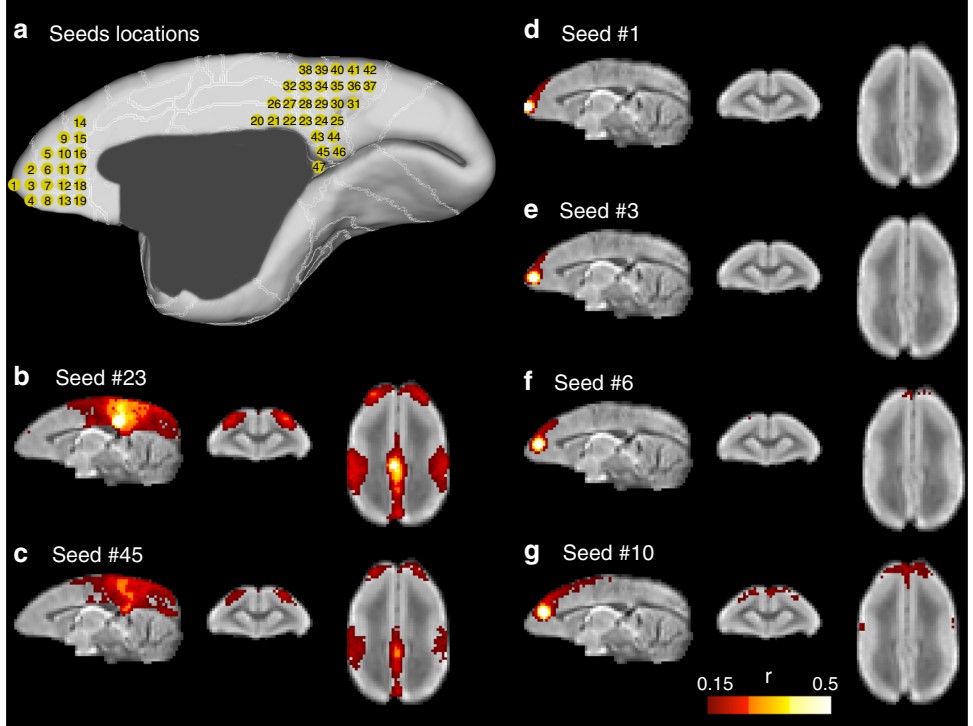

**Fig. 5** Exploratory seed-based analysis. **a** Nineteen seeds were placed across the mPFC and 28 seeds across the medial parietal, posterior cingulate, and retrosplenial cortices. **b–g** Correlation maps of a few selected seeds. Correlation maps of all seeds are listed in Supplementary Fig. 4. For visualization purpose, we set an arbitrary threshold ($r > 0.15$) to remove low correlations. The threshold was selected based on the correlation in the previous ICA, in which the lowest correlation between the medial frontal region and the posterior regions was $r = 0.18$. Correlation maps with a lower threshold ($r > 0.1$) are also provided in Supplementary Fig. 5

network (type-A), according to the positioning of this network in the gradient[17]. Their results suggest the existence of mPFC and frontopolar region as a DMN component in humans, but this region may be too small to be detected as a prominent DMN region in marmosets using fMRI techniques. Another possible explanation for the discrepancy between marmosets and humans may be that the presence of monosynaptic anatomical connections between two different regions does not imply their absolute engagement in a resting-state functional network, and vice-versa.

The DMN of marmosets revealed here is more similar to the human DMN-B, which is characterized by relatively low correlations in the retrosplenial cortex and PHC[16]. However, both human DMN-A and DMN-B have prominent mPFC components, which are missing in the marmoset DMN. The cross-species difference found here also highlights a significant evolutionary difference in DMN: while mPFC occupies a large area and is regarded as a core hub for the DMN in humans, it occupies a small area in macaques, and it cannot be reliably detected in marmosets. This cross-species difference suggests that the mPFC may be less evolutionarily conserved than other DMN regions and that it may be associated with brain functions that are more developed in humans.

In humans, the mPFC is commonly divided into two different components, the dorsomedial PFC (dmPFC) and the ventromedial PFC[1]. Based mainly on evidence from macaque tracing studies, the vmPFC conveys sensory information from orbital PFC to limbic and subcortical structures and is thought to be associated with emotion control and decision making[1,18]. These dense connections between vmPFC (A32v) and orbital PFC also exist in the marmoset brain, and its basic functional role in emotion control and decision making should be preserved in marmosets (Supplementary Fig. 3D). However, in macaques, the mPFC of DMN (A24/32) is located more dorsally than ventrally

according to task-based deactivation (Fig. 2c). Thus, the data obtained from vmPFC in macaques may not fully capture the complex functions of vmPFC in humans, which may be associated not only with emotion control (conserved across primates) but also with functions better developed in humans than in macaques and marmosets.

Although still mostly unknown, the function of dmPFC was studied extensively in human neuroimaging experiments, and this region is believed to be associated with self-relevant mental activities[1,19], which are known to have significant phylogenetic differences across species. For example, the mirror self-recognition (MSR), one of the self-relevant activities, is commonly considered a high intelligence that found only in humans and chimpanzees[20]. A recent study shows that macaques can be trained to pass the MSR test as well[21]. Consistent with their self-relevant mental capacities, the DMN-dmPFC patch can be detected in humans, chimpanzees[4], and macaques[5,6], but humans have the most prominent dmPFC component. As the mPFC is a weak component in the marmoset DMN, self-relevant mental capacities may not be as well developed in marmosets compared to humans.

Directly contrasting the weak mPFC in the marmoset DMN, the dlPFC (peak at areas A8aD and rostral A6DR) appeared to be a major hub of this species DMN (Fig. 2a). The DMN-dlPFC exists in both New World and Old World primates and does not have the dramatic expansion in the human brain that the mPFC does, which suggests that the function of the DMN-dlPFC may be evolutionarily conserved. Both in humans and monkeys, the DMN-dlPFC is located as a transitional area to the premotor cortex, suggesting it may be a polysensory convergence zone. However, its role in brain function demands further investigation. On the one hand, the DMN-dlPFC has been rarely studied or discussed in human neuroimaging studies. On the other hand, the

dlPFC has rarely been investigated as a component of the DMN in NHP. Such investigation may require using other more direct techniques (e.g., electrophysiology), in addition to fMRI.

## Methods

**Animals**. We complied with all relevant ethical regulations for animal testing and research. All procedures in this study were approved by the Animal Care and Use Committee of the National Institute of Neurological Disorders and Stroke. Seven adult marmosets (male, 350–500 g, and 3–9 years old) were recruited for the resting-state fMRI, and two marmosets for the task fMRI (male, 350–450 g, 6 years old). All marmosets were trained and acclimated to laying in the sphinx position in an MRI-compatible cradle, using a protocol described previously[22]. Briefly, in the first week the marmosets were acclimated to the MRI cradle by being held in their natural resting sphinx position in the cradle for incremental time periods starting with 15 min on day 1 and ending with 2 h on day 5. In the second week, the same acclimation period was repeated, this time in the presence of sounds of the typical MRI pulse sequences utilized during MRI and fMRI data acquisition. In the third and final week, individualized 3D-printed helmets that fit each animal's head size and shape were used to immobilize the head of each marmoset. After the 3-week training period, marmosets were fully acclimated to lay in the sphinx position, with their heads comfortably restrained by 3D-printed helmets. This acclimatization procedure is very effective in training the animals to participate in the fMRI studies with their heads restrained comfortably and non-invasively, and with no signs of stress[22].

**Resting-state fMRI data collection**. We collected awake resting-state fMRI data from seven marmosets in a 7 T, 300 mm horizontal magnet (Bruker, Billerica, USA) with 150 mm gradient set capable of generating 450 mT m$^{-1}$ within 150 μs (Resonance Research Inc., Billerica, USA). We designed and used a newly-designed 10-channel phase-array coil[23], which greatly improved the data quality (the spatial resolution was 2× better than our previous study[7]), and thus allowed more accurate analysis of brain networks. Seven marmosets were recruited, and each animal was scanned for 6 runs of resting-state fMRI (one marmoset had 4 runs), using a 2D gradient-echo EPI sequence (TR = 2000 ms, TE = 22.2 ms, flip angle = 70.4°, FOV = 28 × 36 mm, matrix size = 56 × 72, 38 axis slices, resolution = 0.5 mm isotropic, number of averages = 1, 512 time points, each run is 17 min long). Two sets of spin-echo EPI with an opposite phase-encoding direction (LR and RL) were collected for the EPI-distortion correction (TR = 3000 ms, TE = 0.44 ms, flip angle = 90°, FOV = 28 × 36 mm, matrix size = 56 × 72, 38 axis slices, resolution = 0.5 mm isotropic, number of averages = 1 and 8 volumes for each set). After the fMRI session, a rare T$_2$ structural image was also scanned for co-registration purpose (TR = 6000 ms, TE = 9 ms, flip angle = 90°, FOV = 28 × 36 mm, matrix size = 112 × 144, 38 axis slices, resolution = 0.25 × 0.25 × 0.5 mm$^3$, number of averages = 8). All MRI and fMRI data was collected using ParaVision 6.0.1 (Bruker, Billerica, USA).

**Independent component analysis of resting-state fMRI data**. Multi-session temporal concatenation ICA was performed on all data (across subjects and runs; the group-ICA) and on each animal (across runs only; the subject-ICA) using the MELODIC function in FSL[24]. To avoid removing signals of interest, we performed minimal data preprocessing pipeline before ICA, including slice-timing correction using the 3dTshift function in AFNI[25], motion correction using AFNI's 3dvolreg, EPI-distortion correction using FSL's topup[26,27], and registration to a study-specific template. For the registration, each fMRI run was first co-registered to the structural image using rigid-body transformation and the structural image was then nonlinearly transformed to a study-specific template. The fMRI run was registered to the template by concatenating two stages of registration to avoid multiple interpolations. All registrations and template creation were performed using ANTs packages[28]. The preprocessed data were fed into the MELODIC for the ICA analyses. A small spatial smoothing (1 mm FWHM) and a high-pass cut off (100 s) were also performed inside MELODIC. A different number of components (20, 30, and 40) were selected for the ICA estimation. Results and figures showed the estimation of 30 components. The putative DMN component (mainly comprised of dlPFC, PPC, and PCC) can be identified among these ICA components, despite the setting of the number of components. For visualization purpose, a surface of the marmoset cortex was created from the NIH Marmoset Brain Atlas[11] with the Paxinos parcellation[15] using Caret5 software[29]. DMN components were nonlinearly transformed to the template of the NIH Marmoset Brain Atlas and then mapped on to the surface.

**Seed-based functional connectivity analysis**. In addition to the minimal preprocessing pipeline, we created several regressors to remove potential noise and motion effect using AFNI, including demeaned and derivatives of motion parameters, motion-censor regressors (any TR and the previous TR were censored if the motion was >0.2 mm), and band-pass regressors (0.01–0.1 Hz). Seeds were created from the top 100 voxels in each DMN region of the group-ICA component. Time course of each ROI was extracted from the preprocessed data, and functional connectivity (correlation) were calculated between the mean time-courses between any two ROIs.

Besides the ICA-guided seed analysis, we also performed exploratory seed-based analysis on the mPFC, medial parietal, posterior cingulate and retrosplenial cortex. The size of each seed was 27 voxels and the space between the centers of two neighbor seeds was 1 mm. In total, 19 seeds were placed across the mPFC, involving the frontal pole (A10m), A32, A32v, and medial parts of A14R, A9, and A8b. Twenty-eight seeds were placed across the medial parietal, posterior cingulate and retrosplenial cortex, involving A29/30, 23, PGm, 19m, and medial part of V6. We performed whole-brain connectivity analysis on each seed and the average correlation map across all animals was calculated for each seed.

**Task-based fMRI**. Task-based fMRI data were collected from two marmosets in the same magnet with an 8-channel phase-array coil[30], using a 2D gradient-echo EPI sequence (TR = 2000 ms, TE = 26 ms, FOV = 32 × 32 mm, matrix = 64 × 64, slice thickness = 1 mm). The training, task and scanning protocols were described in detail in our previous study[12]. In brief, a block-design paradigm was used (Fig. 1d), which comprised of 16 s long stimuli periods intervened with 20 s long resting periods (gray screen without rewards). Different types of visual stimuli were randomly presented (every 500 ms without gap) for each stimulus block, including marmoset faces, body parts, objects, scrambled images, and fixation points. A marmoset only received sugary liquid rewards when its eye-tracking signal was maintained at the visual stimuli within the defined tolerance range of 5° radius circle. A stimulus block was considered as a valid block only if the eye signal was maintained at the stimuli more than 80% of block duration. Nineteen sessions were performed for one animal (B) with 1715 valid stimulus blocks and 2011 resting blocks in total. Twelve sessions were performed for the other animal (E) with 1211 valid stimulus blocks and 1445 resting blocks.

All data pre-processing was performed using AFNI, including slice-timing correction, motion-correction, spatial smoothing (0.1 mm FWHM), nuisance signal regression (demeaned and derivatives of motion parameters, and motion-censoring regressors with a motion threshold > 0.1 mm). Here, we performed a simple contrast "all valid stimuli block vs. all resting block" to reveal the task-negative regions. We performed analyses on each session separately to obtain statistical maps for each session. Mixed effects meta-analysis was performed on statistical maps across all sessions using the 3dMEMA of AFNI to obtain the final statistical maps, which were thresholded at a voxel-wise threshold of $p < 0.05$, and a cluster-wise threshold of $p < 0.05$ for multiple comparison corrections. The cluster size was estimated by AFNI's 3dClustSim function[25]. The results were mapped onto the surface map mention above for visualization.

**Neuronal tracing database**. We examined the anatomical connectivity of the DMN regions using the Marmoset Brain Architecture Project (Monash version, http://marmoset.braincircuits.org/), which is a retrograde neuronal tracing database that comprises of 141 injections across marmoset brains. As the database used the Paxinos-atlas parcellation scheme, we also used the Paxinos atlas (NIH brain atlas version) to locate our DMN regions on the database. As each DMN patch involved multiple adjacent regions and there may be multiple injections in each region, we presented the tracing map from one of the multiple injections in Fig. 4 and the other related injections in Supplementary Fig. 1.

## Data availability

The resting-state fMRI and the task fMRI data that support the findings of this study are available from the corresponding authors upon reasonable request. The neuronal tracing data are available from the Marmoset Brain Architecture Project (Monash version, http://marmoset.braincircuits.org/).

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

## Acknowledgements
The authors wish to thank Profs. Randy L. Buckner and Marcello Rosa for valuable discussions, Dr. John Newman for proofreading early drafts of this manuscript, Brandon Chen and Madeline Marcelle for their assistance in animal training, and Dr. Soohyun Park for helping with data retrieval of task fMRI stimulation files. The authors are equally grateful to the Scientific and Statistical Computing Core of the NIMH Intramural Research Program for their support with AFNI. This research was supported by the Intramural Research Program of the NIH, NINDS.

## Author contributions
C.L. designed the study, trained animals, collected and analyzed the data, and wrote the manuscript. C.C.-C.Y. collected the data. D.S. trained animals. F.Q.Y. and D.A.L. wrote the manuscript. A.C.S. designed the study and wrote the manuscript.

## Additional information

**Competing interests:** The authors declare no competing interests.

