## [Peer Review File · Nature Communications]

Reviewers' comments:

Reviewer #1 (Remarks to the Author):

This is a very well thought out, and very well written manuscript.

In general, I have few concerns or suggestions.

My primary recommendation would be to modify the the statements regarding the differential weighting of mPFC vs. dIPFC in the marmoset DMN. There is a slight tendency in the current version to want to "have it both ways", with the strong conclusion of a complete shift from mPFC to dIPFC as the rostral hub of the network, whereas the text and the figures support a somewhat more moderate conclusion that the mPFC component is present but much less strongly connected to the posterior hub of the DMN in marmosets vs. humans or Old World monkeys. This point is present in the current MS, and is in no way a barrier to publication, but clarification of the authors' interpretation in the final version would be beneficial.

Minor points:

1) There are a few typos throughout the MS (e.g. "evolutionary" vs "evolutionarily" p.14; line 288).

2) On p.15; lines 308-310, the sentence describing mirrorself recognition contains an editing error "considered as a high intelligence".

Reviewer #2 (Remarks to the Author):

Liu and colleagues present an elegant set of studies to define a marmoset homologue of the default mode network. Building on their previous 2013 fMRI study demonstrating DMN in marmoset, they include additional fMRI studies to show resting-state activity within the homologous network, and demonstrate anatomical connections consistent with the marmoset functional DMN.

This is a well written and well designed study. The findings are a redemonstration their previous putative ICA-based DMN-like network in marmoset. Utilizing an on-off task, the network is now confirmed to be a DMN-homologue, and like higher primates, this network is more active during task-negative conditions. They also investigate the existence of anatomical networks that match the DMN-homologue from retrograde tracer studies in the Marmoset Brain Architecture Project. A key difference from human DMN is the involvement in dorsolateral rather than dorsomedial prefrontal cortex. The anatomical studies do show cortical connections to medialIPFC. They postulate that the failure to see these in the ICA and task studies may be a failure of the fMRI sensitivity. Another explanation for this discrepancy that the authors could discuss may lie in the concordance between anatomically-defined pathways and functional networks. Just because an anatomical connection exists, does not mean it will be engaged in a transient functional network.

Reviewers' comments:

Reviewer #1 (Remarks to the Author):

This is a very well thought out, and very well written manuscript.

In general, I have few concerns or suggestions.

My primary recommendation would be to modify the statements regarding the differential weighting of mPFC vs. dlPFC in the marmoset DMN. There is a slight tendency in the current version to want to "have it both ways", with the strong conclusion of a complete shift from mPFC to dlPFC as the rostral hub of the network, whereas the text and the figures support a somewhat more moderate conclusion that the mPFC component is present but much less strongly connected to the posterior hub of the DMN in marmosets vs. humans or Old World monkeys. This point is present in the current MS, and is in no way a barrier to publication, but clarification of the authors' interpretation in the final version would be beneficial.

Reply: We thank the reviewer for making this important recommendation. In the revised manuscript, we have modified many statements to the more moderate conclusion that the mPFC component is present in the marmoset DMN, but that it is much smaller than the human mPFC, and only weakly connected to the posterior regions of the DMN. The manuscript was revised in many places to tone down on the strong conclusion. We hope that now the text is more consistent throughout the manuscript.

Minor points:

- 1) There are a few typos throughout the MS (e.g. "evolutionary" vs "evolutionarily" p.14; line 288).
- 2) On p.15; lines 308-310, the sentence describing mirrorself recognition contains an editing error "considered as a high intelligence".

Reply: Thank you for your careful reading of our manuscript. We apologize for the typos throughout the manuscript. We have corrected these typos in the revised manuscript.

Reviewer #2 (Remarks to the Author):

Liu and colleagues present an elegant set of studies to define a marmoset homologue of the default mode network. Building on their previous 2013 fMRI study demonstrating DMN in marmoset, they include additional fMRI studies to show resting-state activity within the homologous network, and demonstrate anatomical connections consistent with the marmoset functional DMN.

This is a well written and well designed study. The findings are a redemonstration their previous putative ICA-based DMN-like network in marmoset. Utilizing an on-off task, the network is now confirmed to be a DMN-homologue, and like higher primates, this network is more active during task-negative conditions. They also investigate the

existence of anatomical networks that match the DMN-homologue from retrograde tracer studies in the Marmoset Brain Architecture Project. A key difference from human DMN is the involvement in dorsolateral rather than dorsomedial prefrontal cortex. The anatomical studies do show cortical connections to medialPFC. They postulate that the failure to see these in the ICA and task studies may be a failure of the fMRI sensitivity. Another explanation for this discrepancy that the authors could discuss may lie in the concordance between anatomically-defined pathways and functional networks. Just because an anatomical connection exists, does not mean it will be engaged in a transient functional network.

Reply: We thank the reviewer for the positive comments and for making this very important point. We agree with the reviewer that this is a highly plausible explanation for the discrepancy between functional connectivity and neuronal-tracing studies. In particular, all neuronal-tracing studies have been done with monosynaptic tracers, while functional connectivity is based on complex multi-synaptic anatomical connections. In the revised manuscript, we have added a discussion about possible reasons for the discrepancy on page 14 - lines 21 to 28.

REVIEWERS' COMMENTS:

Reviewer #1 (Remarks to the Author):

The revised manuscript clearly addresses the concerns raised earlier, and I look forward to seeing the final version in print. No further concerns or suggestions.